# Unraveling Factors Affecting Micropropagation of Four Persian Walnut Varieties

**Togzhan Kadylbekovna Yegizbayeva** [1], **Silvia García-García** [2], **Tatyana Viktorovna Yausheva** [1], **Markhabat Kairova** [3], **Amangeldy Kairbekovich Apushev** [4], **Sergey Nikolaevich Oleichenko** [1] and **Ricardo Julian Licea-Moreno** [5,*]

[1]  RSE Issyk State Dendrological Park, Almaty Region 040409, Kazakhstan; togjan26@yandex.ru (T.K.Y.); yausheva_tatyana@mail.ru (T.V.Y.); oleichenko@mail.ru (S.N.O.)
[2]  WalnutRD, 34210 Palencia, Spain; garcia1silvia@gmail.com
[3]  Laboratory of Biotechnology, Botanical Garden, Nur-Sultan City 020000, Kazakhstan; markaigai@mail.ru
[4]  Department of Agronomy, Faculty of Agronomy, Kazakh National Agrarian University, Almaty Region 050000, Kazakhstan; apushev-ak@mail.ru
[5]  Independent Researcher, 04640 Almería, Spain
*   Correspondence: rjliceam@yahoo.es; Tel.: +34-643-767-360

**Abstract:** Walnuts are considered recalcitrant to tissue culture, with a great genetic determinism on all stages of micropropagation; while other factors, also with great impact, become more complicated with the reproduction of newly realized varieties. In this research, a holistic approach aimed to determine the influence of genotype and the nutritive formulation throughout the whole cycle of micropropagation of four Persian walnut varieties (*Juglans regia* L.) was presented. During the in vitro establishment it was determined that besides genotype and culture medium, the effect of collection season and the likely interaction amongst factors had a great influence on the successful introduction of all four genotypes. However, all cultures were affected by a deep decay, being necessary the introduction of ethylenediamine di-2-hydroxyphenyl acetate ferric, as iron source, and Phloroglucinol in both Murashige and Skoog (1962) and the corrected Driver and Kuniyuki (1987) formulations. These modifications allowed the stabilization of cultures, maintaining thereafter a steady quality. Either proliferation, rooting and ex vitro survival of four clones were affected by the culture medium, obtaining the best results with the corrected Driver and Kuniyuki (1987) formulation. Finally, in vitro plants produced from all clones were acclimated with high survival rates (75.9–91.1% for the best culture medium), depending of clone and the culture medium used. Microsatellite analysis showed that micropropagated plants maintained the same genetic profiles of their corresponding donor trees. These results might contribute to deepening of the understanding of factors that determine the success of micropropagation of walnuts, and the extents of its influence; whereas, it sets the basis for the commercial micropropagation of all four clones.

**Keywords:** *Juglans*; reproducibility; proliferation rate; rooting; recalcitrant; Phloroglucinol; FeED-DHA

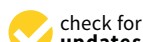



## 1. Introduction

Persian or English walnut (*Juglans regia* L.) is a valuable species from Juglandaceae family, highly appreciated for its edible nuts and wood. It is native to Central Asia, growing wild or semi-cultivated from the Balkans, passing through Turkey, China and Eastern Himalaya [1–3]. Genetic studies have suggested that section Juglans, being *J. regia* L. its only member, might has evolved isolated in Central Asia [4], extending from Xinjiang province (China), parts of Kazakhstan, Uzbekistan and Kyrgyzstan, and from Central Himalaya and Iran to Western Asia, and later to Eastern Europe [5].

Walnut has become a globally cultivated crop that, besides the increasing exploitations throughout its traditional culture areas in North America, Europe and Asia, has also been

spread out to the Southern hemisphere [1]. Although the main nut producers world-wide are USA, China and Turkey, the increasing interest and commercialization of walnut products have pushed several countries from Central Asia to initiate their own genetic programs, allowing the exploitation of the great natural biodiversity that they harbor [6–9]. Kazakhstan is in the Northern line of walnut cultivation, occupying a strategic position, close to the important sale markets as Russia, China, India and the Arabian countries. However, the scarcity of commercial varieties adapted to its harsh climate conditions has hindered the spread of walnut exploitations. While the cultivation of well-known commercial genotypes has failed in the Southern regions of Kazakhstan, local genotypes have been selected for their high resistance to freezing temperatures. Besides the exploitation of its genetic resources, some varieties have also been introduced from China and Uzbekistan, which have showed high production and adaptability to the local conditions. Therefore, the next step would be to provide nurseries with sufficient high-quality plant materials of the new selections and clones, as the basis for the establishment of productive plantations, as well as to perform adaptability studies.

Undoubtedly, only tissue culture would offer the possibility to produce high volumes of certified materials in short time periods, maintaining the genetic identity of propagated clones. Although several authors have published micropropagation protocols for various genotypes [10–17], walnuts are considered recalcitrant to tissue culture, becoming difficult the in vitro propagation of the newly realized genotypes. Additionally, the low reproducibility of micropropagation protocols is another critical factor that makes the reproduction of some varieties a more complex process; forcing to perform their adaptation on a genotype-to-genotype base, and laboratory-to-laboratory as well. It is known that the results of walnut micropropagation are highly dependent on genotype; however, the use of the same protocol under different conditions, even performed by similar staff, might cause important variations, as has been observed for rooting of the variety 'Chandler' [14,18].

Along with genotype, the nutritive formulation has a great influence on all stages of micropropagation. Nowadays the Driver and Kuniyuki (DKW) culture medium [10] has been the most used formulation for tissue culture of walnuts. However, some researchers have reported better results using Murashige and Skoog (MS) formulation [19] in all stages of micropropagation [13,15] or only in some steps [18,20,21]. This suggests that changing nutritive requirements might arise during the different stages of micropropagation, besides great variations could be expected between both formulations. Nevertheless, to the best of our knowledge, DKW and MS formulations have never been compared under the same conditions throughout all micropropagation cycles of species from Juglandaceae.

Therefore, in this research the effects of formulations MS and DKW on the different stages of micropropagation of four Persian walnut genotypes were assessed. Additionally, the influence of two different collection season of starting material on in vitro establishment was also analyzed. Hence, the goals of research were (1) to evaluate some factors with great impact on the successful micropropagation of four Persian walnut genotypes, that determine (2) the creation of stable in vitro cultures, establishing thus the bases for conservation of clones, and for their commercial micropropagation.

## 2. Materials and Methods

### 2.1. Plant Material

Four Persian walnut (*J. regia* L.) genotypes for nut production were used. All four genotypes are included in an ongoing improvement program conducted by RSE Issyk State Dendrological Park (ISDP), Ministry of Ecology, Geology and Natural Resources of Republic of Kazakhstan. These genotypes, besides producing sustained high-quality yields, have been selected for their resistance to freezing temperatures, becoming useful for regions characterized by freezing temperatures occurring during early fall and late spring. 'Form 3' and 'Form 4' were selected in relict stands located on the territory of the Sairam-Ugam State National Natural Park (41°55′46.0″ N 69°57′18.2″ E), Kazakhstan. Whereas, 'Ideal' is a variety from Uzbekistan, traditionally used in small exploitations in Shymkent region

(Kazakhstan), and 'Liaohe-1' is an apomictic Chinese variety [22]. For in vitro introductions, sticks from varieties 'Ideal', 'Form 3', and 'Form 4' were obtained from the ISDP arboretum (43°27′18.7″ N 77°27′16.5″ E), and variety 'Liaohe-1' was provided by Caspian Food LLP (41°26′11.6″ N 69°05′25.9″ E), Turkestan region, Kazakhstan. Taxonomic identification was carried out at ISDP using descriptors from the International Union for the Protection of New Varieties of Plants (UPOV). Healthy and vigorous grafted adult trees (12 years) were selected as starting materials.

### 2.2. Culture Conditions

The corrected formulations of DKW (DKWC) [11] and MS [19] were used as culture media. DKWC was prepared from stock solutions, and for MS a dry powder (code 5519, Sigma Aldrich; St. Louis, MO, USA) was used, except for root expression, for which stocks solutions were also prepared. The pH was adjusted to 5.7 with NaOH (0.1 N); afterward the culture media were sterilized by autoclaving during 20 min at 121 °C. As critical steps toward commercial micropropagation, proliferation and rooting stages were performed in parallel in two different laboratories. These were a laboratory from ISDP, from now on laboratory 1, and a laboratory from WalnutRD (Palencia; Spain), a highly experienced laboratory in walnut micropropagation, as well as other forestall species, from now on laboratory 2. Although the same protocol was followed as much as was possible, some variations occurred. Whereas the same photoperiod (16/8 h) was used in both laboratories, there were met differences in the light intensities and the temperatures. Thus, in laboratory 1, there was mean light intensity of 3500 lx (PH LED tube 1200 mm 2*36 W) and temperatures from 24 to 26 °C; while temperatures of culture rooms in laboratory 2 were maintained at 24 ± 2 °C, with an average photosynthetic photon flux density (PPFD) of 50 μmol m$^{-2}$ s$^{-1}$ (≈4000 lx). Chemicals in laboratory 1 were provided for PanReac AppliChem ITW Reagents (Barcelona; Spain) and Duchefa Biochemie (Haarlem; The Netherlands) was the supplier for laboratory 2. For in vitro introduction, proliferation, and root pre-induction, culture media were gelled with plant agar (code A2111, PanReac AppliChem ITW Reagents; Barcelona; Spain) in laboratory 1 (7 g/L), and with industrial agar (code 1804, Condalab; Torrejon de Ardoz; Spain) in laboratory 2 (5.5 g/L). In both laboratories, rooting was performed in two steps: root induction, in total darkness (5–7 days), and root expression, under a 16/8 h photoperiod. For this stage healthy and vigorous microshoots from multiplication, without any sign of defoliation and/or wilting, and at least 20 mm length were used, as had been recommended for clones of the walnut hybrid progeny Mj209xRa [17,23].

### 2.3. In Vitro Introductions

In vitro introductions were only performed in laboratory 1 during 2018. Two procedures were followed, (1) introductions from sticks bearing dormant buds collected in February 2018 but formed in 2017, and (2) introductions of softwood branches sprouted in 2018, collected in May (of the same year). Materials were individually labeled and transferred to the laboratory within 12 h after being collected. Cuttings were profusely washed with sterilized water and household soap and washed again to remove the soap residues. For procedure 1, cuttings were placed in cups (1000 mL) with water and incubated in a culture room under a photoperiod 16/8 h to promote budding. Water was changed every 3 days and the base of sticks were removed, resembling the recommendations for Mj209xRa hybrids [17]. Sprouted shoots with >2 cm length were used for the next step. Whereas, sticks collected in May (procedure 2) were cut before disinfection in segments bearing one node. Afterwards, the segments were treated with a solution of Triclosan (Billio Chemistry LLC, Novokuznetsk, Kemerovo region, Russia) during 1 h in an orbital shaker at 200 rpm. On top of this pre-cleaning procedure, explants were disinfected soaking them for 3 min in a solution of HgCl$_2$ (0.1%) with the addition of Tween-20; followed by intensive washing with sterilized water on an orbital shaker during 5, 10, and 15 min. Explants were individually inoculated in glass test tubes (150 × 20 mm) containing 10 mL of either DKWC or MS

culture medium supplemented with 4.4 µM of 6-bezylaminopurine (BAP) and 0.05 µM of indole-3-butyric acid (IBA) and maintained under the standard photoperiod conditions. Contaminated or dead explants as well as those with profuse phenolic releasing were counted and discarded. Only were considered those treatments with at least 10 explants.

### 2.4. Proliferation

Laboratory 1: Initially, glass jars (200 cm$^3$, 100 mm height and 60 mm diameter) were used. Later, cultures were inoculated in polypropylene containers (O118/120+OD118, Microbox, SacO$_2$, Deinze; Belgium). As for the in vitro establishment, DKWC and MS formulations were supplemented with the same BAP and IBA concentrations used for in vitro introductions. In this stage was necessary the introduction of Phloroglucinol (0.4 mM, PG) and the replacement of ferric ethylenediaminetetraacetic acid (FeEDTA) by ethylenediamine di-2-hydroxyphenyl acetate ferric (119 mg/L, FeEDDHA), as has been recommended for micropropagation of Mj209xRa clones [17,23], naming culture media DKWC-m and MS-m. Ten explants per vessel were inoculated in 100 mL of culture medium. Subcultures were performed during the 5th week. The number of nodes per microshoot, the length of the microshoots, and the diameter of callus (measured for its widest part) were evaluated for each treatment at the end of subculture.

Laboratory 2: Explants from the four genotypes were provided by ISDP's laboratory in March 2019. After receiving the plant materials, these were individually subcultured to fresh medium in test tubes (150 × 25 mm) to control the appearance of microbial contaminants. During the first three subcultures no further actions were performed, more than the increase of volume of materials as well as bringing them to their stabilization. For proliferation, the same model of polypropylene's containers was used. Only DKWC-m formulation was used. Ten explants were inoculated per container with 100 mL of culture medium, as in laboratory 1. Subcultures were performed every 6 weeks. The number of nodes per microshoot were counted at the end of each subculture.

### 2.5. Rooting

Laboratory 1: The procedures described for Persian walnut [24], with some variations, and Mj209xRa clones [17], using MS and DKWC formulations, respectively, were followed, introducing some modifications. Thus, MS ($\frac{1}{4}$ macronutrients) was supplemented with IBA (14.7 µM), whereas 50 µM of IBA was added to DKWC ($\frac{1}{2}$ macronutrients) culture medium. Regardless of culture medium, FeEDDHA was used as iron source instead of FeEDTA. For both formulations, liquid culture media and vermiculite (commercial brand LETTO, LLC "Biotek", Miass, Chelyabinsk region, Russia), without IBA, were used for root formation. During the 4th week, rooted microshoots and number of roots per microshoot were counted, and the general state of rooted microshoots per treatment was registered.

Laboratory 2: Only the procedure described for Mj209xRa clones [17] was followed. Thus, cultures were incubated in the darkness in a culture medium with macronutrients of DKWC reduced to 50% and 50 µM of IBA. Afterwards, microshoots were transferred to the expression sub-stage in the same culture media with vermiculite (type 3, Projar, Valencia; Spain), without IBA nor agar. After 4 weeks, the number of rooted microshoots per genotype were counted, and the general state of rooted microshoots was annotated.

### 2.6. Acclimatization

The weaning was performed in two steps. Initially, rooted microshoots were planted in peat tablets (Jiffy-7, 41 mm diameter, product code 32170138, Oslo, Norway) and incubated in mini-greenhouses under controlled conditions. Mini greenhouses were covered with glass to maintain high relative humidity ($\approx 90 \pm 2\%$), and the substrate was lightly watered once per day during the first week. Thereafter to the 3rd week, glass covers were removed slowly, and micropropagated plants were watered twice per week, in such a way that plants were acclimated step-by-step to a gradual-reduced humidity. In this stage a photoperiod of 16/8 h was used, with an average light intensity of 3500 lx; whereas temperature was

24–26 °C. During the 4th week, surviving plants were transplanted to plastic cups (500 cm$^3$, upper diameter 93 mm, lower diameter 55 mm, height 135 mm) with a mixture (rate 2:1) of peat (Suliflor SF2, Radviliškis, Lithuania) and vermiculite (LLC "Biotek", Miass, Chelyabinsk region, Russia). Pots were placed in a greenhouse under normal conditions, with oscillating temperatures and relative humidity ranging from 17 to 32 °C and from 45 to 75%, respectively, and an average maximum light intensity of 4600 lx. Plants were watered 1 to 2 times per week, depending on the weather and the state of substrate. When micropropagated plants were weaned (6th week), survival and quality of plants of each genotype from both culture media were counted, and the general state was annotated.

### 2.7. Analysis of Genetic Stability of Micropropagated Clones

To assess the genetic stability leaves of donor trees, and their corresponding micropropagated plants were used. Five microsatellite loci (WGA001, WGA009, WGA089, WGA276, and WGA321 [25]) used to differentiate varieties of *J. regia* were selected. Genomic DNA was extracted using DNeasy Plant Mini Kit (Qiagen, Hilden, Germany). DNA quality and quantification were assessed in 0.9% of agarose (Amresco, Solon, OH, USA) and by spectrophotometry (Nanodrop ND-1000, ThermoScientific), respectively. PCR reactions were carried out in a final volume of 25 μL, consisting of 1× PCR buffer (20 mM Tris- Cl pH 8.4, 50 mM KCl, ThermoScientific), 2.0 mM MgCl$_2$, 0.2 mM dNTPs (ThermoScientific), 0.2 μM of each primer (Sigma-Aldrich, Germany), 0.25 units of Taq-DNA recombinant polymerase (ThermoScientific) and 25 ng of genomic DNA. PCR amplifications consisted of 5 min for initial denaturation at 94 °C, 35 cycles of 45 s at 94 °C, 45 s at the annealing temperature specific to each pair primer, and 45 s at 72 °C, and a final step for 10 min at 72 °C. For allele assignation, PCR products were fractioned in agarose gel (3.0%, MetaPhor, Lonza) at 120 V during 60 min. The GeneRuler 100 bp (ThermoScientific) and 50 bp ladder (Invitrogen, Lithuania) were used as the DNA length reference. PCR amplicons were compared to the DNA ladder by gel scanner (GelDoc, BioRad, Hercules, CA, USA).

### 2.8. Experimental Design and Statistical Data Processing

A randomized model was applied for all experiments. The container was the basic experimental unit; therefore, the average of the containers was used as individual data for analysis. Each treatment was composed of at least three experimental units for all experiments. During the proliferation (stage II), at least three subcultures were assessed, being considered each subculture a repetition. The analyses of variances (ANOVA) were performed with R [26]. Normal distribution and homogeneity of data were determined using Shapiro–Wilks' and Levene's tests, respectively. When necessary, LSD post hoc and Pearson's (data normally distributed) or Spearman (data non normally distributed) correlation tests were performed. Those variables that did not meet the ANOVA's assumptions were analyzed with a Kruskal–Wallis test.

## 3. Results and Discussion

### 3.1. In Vitro Introduction and Establishment

Establishment is likely the most critical and unpredictable stage of walnut micropropagation, especially when somatic organs from field-growing trees are used as starting materials [11,20,27]. Thus, noteworthy was the successful in vitro establishment of all four genotypes. From the bulk of 765 explants that were introduced, 448 (58.6%) overcame the initial phase of in vitro establishment. The rest of the explants were lost due to bacterial (19.9%) and fungal (11.1%) contaminations, as well as by the profuse and uncontrolled phenolics releasing (10.5%), which caused their death. However, striking differences for all the assessed variables were observed amongst genotypes, culture medium formulations and the phenological state of donor trees.

Probably the most obvious and expected result was the differences observed amongst clones (Figure 1). From the earliest reports it is a well-known variation factor for pure walnut species [16,28], and hybrids [12,17]. Thus, the establishment goes from 0%, for

'Ideal' clone in DKWC culture medium using FeEDDHA, to 100% of success, for explants from 'Liaohe-1' introduced in May, cultured on DKWC supplemented with FeEDDHA (Figure 1c).

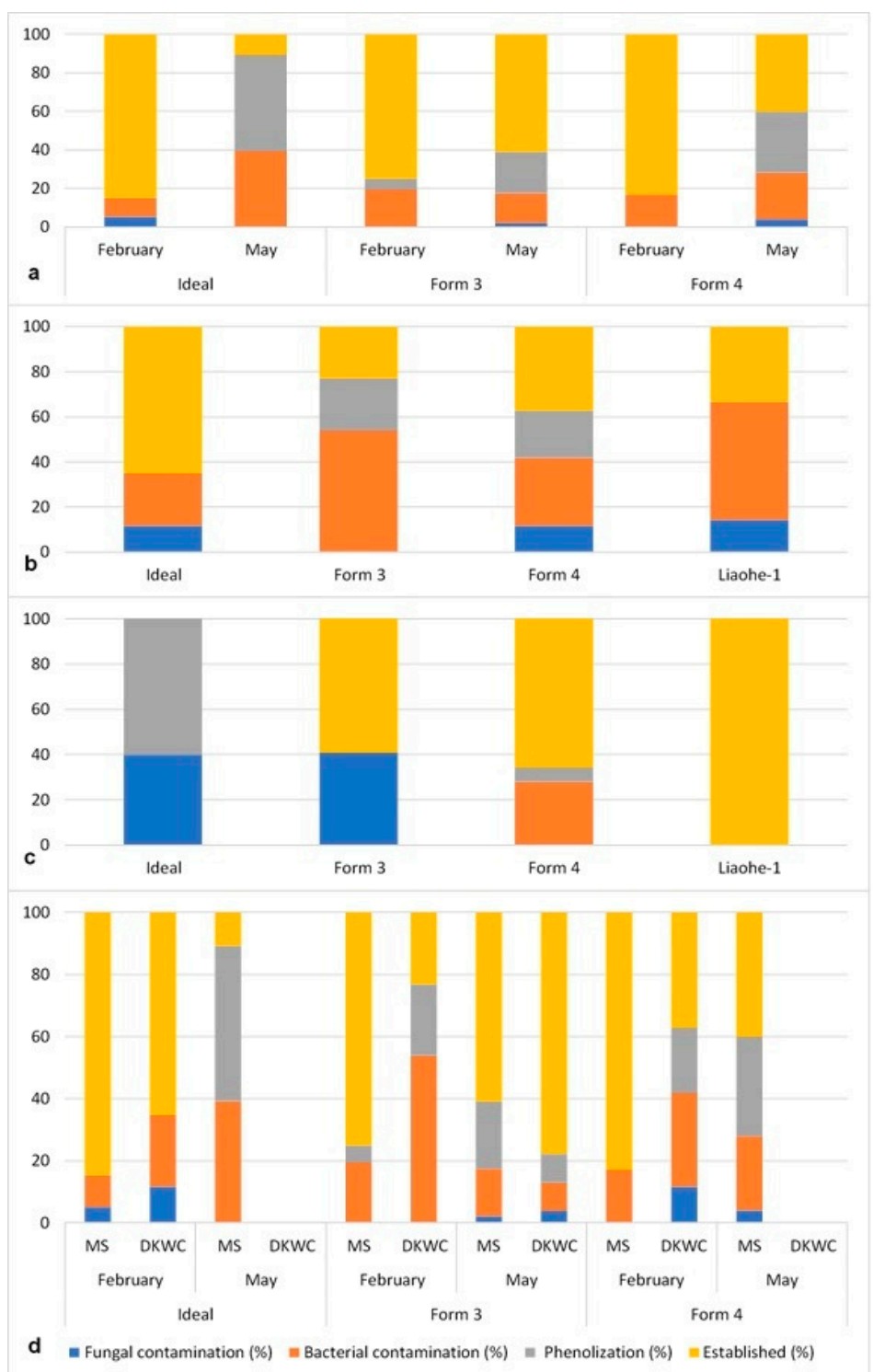

**Figure 1.** Results of in vitro introduction and effects of different factors on the success of establishment. (**a**) Effects of introduction season, and genotype using the MS culture medium. (**b**) Effects of genotype on February's introductions using the DKWC formulation. (**c**) Effects of genotype on May's introductions using the DKWC formulation supplemented with FeEDDHA instead FeEDTA. (**d**) Combined effects of genotype, introduction season and culture medium.

Contamination losses were determined by the genotype, the culture medium, and the introduction season (Figure 1), as well as by the likely interaction amongst factors (Figure 1d). Gruselle and Boxus [20] also registered different infection percentages amongst three pure walnut species and the hybrid Paradox, the percentage of contaminated explants ranging between 0 and 67.8%. Microorganisms are ubiquitous from eukaryotes, including plants, acting as endo and ectosymbionts, most of them essential for their life [29], therefore every genotype might bring its own microbiome. At the same time, this microbiome might change, influenced by many factors such as season, location and age, even amongst the different organs of plants [30]. It has been found that bacterial communities inhabiting the perennial wild mustard *Boechera stricta* vary with locations, the age of plants, and genotypes, also registering variations amongst the different organs of plants [31]. Besides, although some bacteria seem common to several tree species, great changes could be observed in their population size [32]. Therefore, the results here presented may be influenced by potential seasonal variations in microbiological diversity of donor trees, as well as for the different in vitro culture conditions used, i.e., culture medium. Hence, while for clone 'Form 3' the bacterial contaminants were more prevalent for winter introductions, explants from 'Ideal' inoculated in culture medium MS showed more affectations during May. In general, microbial contaminations affected more explants in DKWC than in MS culture medium (Figure 1d), except for May's introductions of clone 'Liaohe-1' (Figure 1c).

Great contrasts were also observed for phenolic releasing. Hence, introductions from May were more affected by phenolization than those performed in February. The great vegetative activity that donor trees show during the mid-spring might drive the profuse exudation of phenolics to culture media. Solar et al. [33] have registered similar time course profiles for accumulation of phenolic substances of several Persian walnut varieties, being increased the production of flavonoids from the beginning of the growth cycle in May to the end of August. Hence, a large reduction of phenolics production and the number of dead explants has been registered for walnut hybrids and Persian walnut introduced during winter [34,35]. As genotype cannot be controlled, choice of the right moment of introduction might increase the success of in vitro establishment. Thus, 'Ideal', 'Form 3' and 'Form 4' clones were highly affected by phenolization, although its extent was also highly dependent on the recollection season and the culture medium used (Figure 1). Interestingly the 'Liaohe-1' clone was not affected at all for phenolic releasing, despite the collection season, being probably the cause of obtaining the highest establishment success, ranging from 33.3 to 100%, with an average of 74.5%. This might agree with the findings of Solar et al. [33] and Pereira et al. [36], that have registered for *J. regia* that the production of phenols is determined by the genotype.

To a lesser extent, but also important, the culture medium had certain influence on phenolics releasing (Figure 1d). Whereas MS formulation seems better than DKWC for winter introductions, during the spring, explants from 'Form 3' realized less phenolics when they were cultured in DKWC than in MS (Figure 1d). Although DKW is the most used formulation for in vitro culture of walnuts, MS formulation has rendered suitable results for some specific stages of micropropagation, as for in vitro establishment [20,21]. Regarding the control of phenolization, some authors have expressed contradictory results. While Revilla et al. [37] used DKW to reduce the exudation of Persian walnut explants in MS culture medium, Lone et al. [21] obtained important reductions of browning using MS formulation. These results suggest the need to consider the different factors that might determine the success, or failure, of in vitro establishment, once dramatic outcomes might arise. Sometimes few opportunities are available or reduced quantities of materials are accessible to approach the introduction of valuable genotypes, therefore, besides the expertise of technicians, the correct timing, planning and management might increase the possibilities to succeed with in vitro establishment.

### 3.2. Proliferation

After the establishment, cultures showed a general decay not associated to microbial contaminations, especially those in DKWC culture medium. This behavior has been previously reported for several walnut species [28,38,39], even Driver and Kuniyuki [10] have indicated that some deficiencies may become visible with DKW. The signs of declination of cultures were showed as a sudden reduction of vigor of microshoots, rendering low multiplication rates, and a deep yellowish color of leaves (Figure 2a). The loss of vigor has been described amongst the most important causes of failure of in vitro culture of walnut, actually it was the main reason for creating the DKW formulation [10]. To this point, microshoots were also characterized by the small size, or the lack, of their basal calli; causing likely the progressive diminishing of growth, as has been demonstrated for Mj209xRa clones [17,23]. It should be also considered that the loss of vigor is especially conspicuous using adult trees as donors for start of the in vitro introductions, as was observed for clones of American black walnut [40]. However, the introduction of some modifications in both culture media promoted the stabilization of cultures. These were the supplementing of culture media with PG, and the replacement of FeEFDTA by FeEDDHA. Some recent studies have demonstrated the positive effects on micropropagation of several plant species both PG [17,23,41–43] and FeEDDHA [17,23,44–48]. Therefore, formulations with these changes were named MS-m and DKWC-m.

Formulation was the main variation factor during proliferation (Table 1). Despite genotype, microshoots grew more in DKWC-m culture medium than in MS-m, being even folded the length for clones 'Form 3' and 'Ideal' (Table 2). Although less evident, the multiplication rates were also increased significantly in DKWC-m formulation, suggesting that growth occurred mainly through the elongation of internodal spaces rather than for the formation of more nodes. Additionally, noteworthy was the size of calli formed in DKWC-m culture medium, surpassing up to 4.4 times the diameter of those in MS-m formulation (Table 2). These results agree with those of Driver and Kuniyuki [10], who found that multiplication in DKW was promoted four times more than in B5, Cheng, MS and WPM culture media; also reporting the improvement of quality of microshoots. Although DKW formulation has been used predominantly for walnut micropropagation, other authors have registered similar results for cultures in MS, even better than those observed in the DKW medium. Thus, axillary buds of *J. cinerea* were more elongated in MS than in DKW [13]. Whereas, the length of microshoots of American black walnut were similar in both formulations; although, the percentage of elongated explants was lower in MS medium, producing also more hyperhydric microshoots than in DKW [48]. In any case, the lack of interaction between clones and culture media for all the assessed variables (Table 1) point out that DKWC-m was better than MS-m for the growth of all genotypes here investigated.

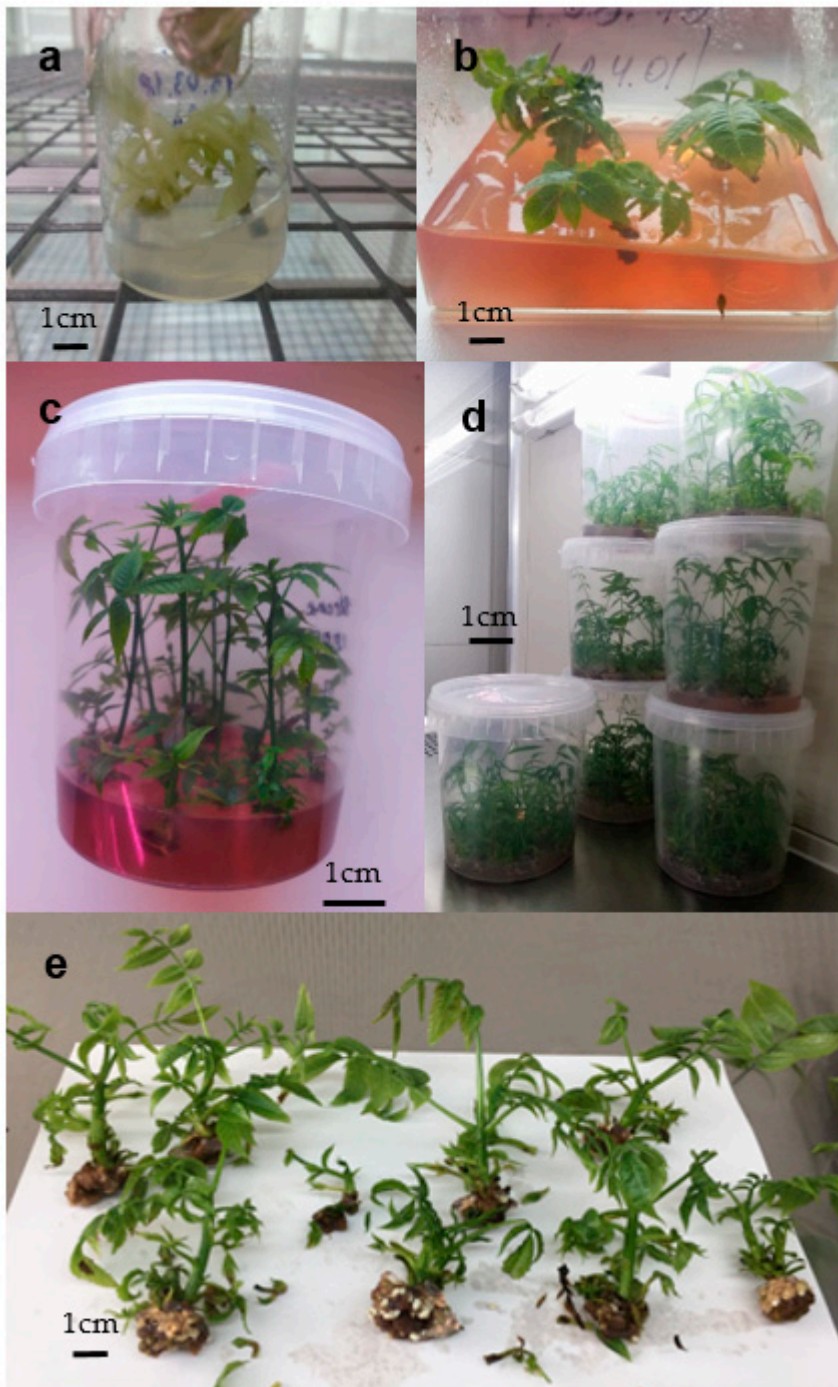

**Figure 2.** Microshoots in the stage II (proliferation) cultivated in laboratories 1 (**a–d**) and 2 (**e**) with the corrected formulation of DKWC [19]. (**a**) Microshoots cultured in the DKWC formulation using FeEDTA. (**b**) Microshoots cultured in the DKWC formulation using FeEDDHA. (**c–e**) Microshoots cultured in the modified formulation of DKWC (DKWC-m) supplemented with Phloroglucinol, and using FeEDDHA as iron source.

**Table 1.** Resume of ANOVAs for each variation factor, including the main interactions, and the assessed variables during proliferation, rooting and acclimatization.

| | Variation Factor | | | | |
|---|---|---|---|---|---|
| Variable | Genotype (G) | Culture Medium (CM) | Laboratory [a] (Lab) | GxCM | GxLab [a] |
| Length of stem | ** | *** | nd | ns | nd |
| Proliferation rate | *** | *** | *** | ns | ns |
| Diameter of calli | ** | *** | nd | ns | nd |
| Rooting [b] | *** | *** | *** | nd | nd |
| Roots/microshoot | ns | *** | *** | ns | ns |
| Survival | ** | *** | nd | ns | nd |

[a] For statistical analysis only data from DKWC formulation were included. [b] Since rooting data do not fit to ANOVA assumptions, a Kruskal–Wallis test was performed. nd: not determined; ns: non-significant. ** signification $p \leq 0.01$ *** signification $p \leq 0.001$.

**Table 2.** Effects of genotype and culture medium on proliferation, rooting and survival in laboratories 1 (ISDP) and 2 (WalnutRD).

| Genotype | Lab | Culture Medium | Length of Stem (mm) | Proliferation Rates | Diameter of Stem (mm) | Rooting (%) | Roots/Microshoot | Survival (%) |
|---|---|---|---|---|---|---|---|---|
| Form 3 | 1 | MS-m | 15.4 ± 4.2 [a] | 2.0 ± 0.9 [a] | 3.2 ± 0.7 [a] | 56.7 ± 5.8 [a] | 2.3 ± 1.6 [ab] | 64.4 ± 3.8 [ab] |
| | | DKWC-m | 33.3 ± 4.4 [d] | 3.0 ± 0.1 [bc/B] | 14.2 ± 2.1 [c] | 76.7 ± 5.8 [b/C] | 3.6 ± 1.0 [cd/B] | 82.7 ± 6.8 [cd] |
| | 2 | DKWC-m | - | 2.7 ± 0.8 [B] | - | 54.0 ± 6.9 [B] | 3.5 ± 1.2 [B] | - |
| Form 4 | 1 | MS-m | 21.2 ± 5.4 [b] | 2.3 ± 0.6 [a] | 3.9 ± 1.5 [ab] | 76.7 ± 5.8 [b] | 2.2 ± 1.1 [a] | 52.4 ± 4.1 [a] |
| | | DKWC-m | 36.6 ± 3.7 [e] | 3.0 ± 0.1 [bc/B] | 16.5 ± 1.5 [d] | 73.3 ± 5.8 [b/C] | 3.7 ± 1.6 [cd/B] | 82.1 ± 15.6 [cd] |
| | 2 | DKWC-m | - | 2.3 ± 0.5 [A] | - | 42.0 ± 6.9 [AB] | 2.0 ± 1.1 [A] | - |
| Ideal | 1 | MS-m | 17.5 ± 2.8 [a] | 2.7 ± 1.3 [b] | 4.4 ± 1.3 [b] | 60.0 ± 10.0 [ab] | 3.1 ± 0.8 [bc] | 49.1 ± 8.6 [a] |
| | | DKWC-m | 39.1 ± 6.3 [f] | 3.6 ± 0.5 [de/C] | 17.3 ± 2.4 [de] | 70.0 ± 10.0 [b/C] | 3.8 ± 1.4 [cd/B] | 75.9 ± 16.5 [bcd] |
| | 2 | DKWC-m | - | 2.2 ± 0.7 [A] | - | 32.0 ± 6.9 [A] | 2.1 ± 0.9 [A] | - |
| Liaohe-1 | 1 | MS-m | 23.9 ± 3.1 [c] | 3.2 ± 0.9 [cd] | 4.2 ± 1.5 [b] | 56.7 ± 5.8 [a] | 3.1 ± 1.1 [bc] | 71.1 ± 7.7 [bc] |
| | | DKWC-m | 45.6 ± 7.0 [g] | 3.7 ± 0.5 [e/C] | 18.0 ± 2.3 [e] | 66.7 ± 15.3 [ab] | 4.1 ± 1.4 [d] | 91.1 ± 7.8 [d] |
| | 2 | DKWC-m | - | 2.0 ± 0.8 [A] | - | 0 | 0 | - |

Values with the same letter are not statistically different. Significance in lower case letters is only referred to comparison between MS-m and DKWC-m in laboratory 1. Significance in capital letters is only referred to DKWC-m in both laboratories. LSD test ($p \leq 0.05$). Rooting percentage was analyzed with Kruskal–Wallis test ($p \leq 0.05$).

Like the establishment, for proliferation it was also observed that the more vigorous microshoots had big basal calli, and vice versa. The basal callus is an unorganized structure that those vigorous and healthy microshoots bear. Little is known about its role in walnut micropropagation. Some authors have suggested that it would be convenient to reduce the size of calli [10]; while, in other investigations these have been associated with the growth of microshoots [39]. Thus, results obtained by Sharifian et al. [49] showed that there is some relationship between the size of basal callus and the growth of three Persian walnut varieties. Similarly, for Mj209xRa clones a high correlation between these variables has been determined [17]. Like these authors, regardless the genotype and the culture medium used, a high correlation between callus size and the length of microshoots (Pearson coefficient r = 0.92, $p < 0.0001$, n = 240) was calculated; even an important relationship with the multiplication rates was determined (Pearson coefficient r = 0.49, $p < 0.0001$, n = 240). Although the size of calli cannot be used as an accurate proliferation predictor, it might become an important indicator of the state of cultures.

As a key step toward the commercial reproduction of clones, the same protocol [17] was reproduced in a second laboratory. All genotypes reacted fast and quite well to the new conditions, allowing to proceed with evaluations after the 4th proliferation cycle. As for laboratory 1, there were differences amongst clones (Table 2); however, the most

outstanding result was the lack of statistical significance for the interaction genotype-laboratory (Table 1). Thus, although the highest multiplication rates were obtained in laboratory 1, in laboratory 2 microshoots of high quality were also steadily produced (Figure 2d).

Since the commercial scale up of experimental protocol might imply the introduction of variations, some of them uncontrollable, it would be desirable to anticipate the assessment of its degree of reproducibility. Here is presented an approach to this complex process. Although many factors were exactly copied in both places (genotype, culture medium, kind of vessel, kind of explant, inoculum density, amongst the most important), apparently small variations (temperature, light intensity, vessels per shelves, organization of vessels in shelves, amongst the most important), even critical elements as human factors and chemical brand, occurred; likely causing the observed variations. While great variations in protocols might provoke the complete failure of cultures, the accumulations of apparent small changes, more difficult to control, might also drive important differences, them becoming equally critical [50]. Regarding this, Piqueras and Debergh [51] have stated that many factors are responsible for changes in the morphology of micropropagated plants; while some are very obvious (plant growth regulators), others are more surprising and often not given due consideration or even ignored (gas phase, container type, place on a shelf, etc.). Nevertheless, despite differences, both laboratories were able to produce high quality and rootable microshoots.

*3.3. Rooting*

Previous screening (data not showed) allowed to determine the interaction between culture medium and IBA concentration on rooting. Thus, for MS formulation the best results were obtained with 14.7 μM of IBA. Whereas for DKWC, the highest root formations were always observed using 50 μM; even microshoots from 'Form 3' failed completely to form roots with lower IBA (14.7 μM) concentrations.

Like proliferation, rooting was determined by genotype and culture medium; although the interaction between clones and formulations did not exist (Table 1). Regardless of the differences registered for rooting, these were less evident than the growth of microshoots during proliferation, except for clone 'Form 3' (Table 2). Meanwhile, clearer were the differences for the number of roots per microshoot, significantly exceeding that obtained in DKWC to those from culture medium with MS formulation. The quality of microshoots at the end of root expression was also higher when these were cultured in DWKC variant (Figure 3a); producing the microshoots from MS shorter roots than in DKWC. Additionally, microshoots from MS culture medium were more affected by defoliation than those from DKWC (Figure 3b), which might represent a disadvantage during hardening, as has been observed by McGranahan et al. [52] and Licea-Moreno [34]. During the 80's, MS was widely used for walnut micropropagation [37,39,53], being replaced by DKW after its creation [10]. Several authors have reported for different walnut genotypes better rooting success using DKW than MS [20,54]. There are plenty of comparisons amongst different formulations on growth and rooting of several genotypes in the literature; however, conclusions regarding the superiority of a culture medium over the other are greatly dependent on genotypes, and likely on the particular culture conditions. Thus, different culture media for the different micropropagation stages are frequently used. For several walnut species and hybrids, Gruselle and Boxus [20] although they used MS formulation during the first stages of micropropagation, for rooting it was replaced by DKW. On the contrary, Navatel and Bourrain [15–24] used DKW for establishment and proliferation, and MS for root formation. Even, different nutritive formulations have been used in the different substages of walnut rooting [11,18,55,56] obtaining different outcomes.

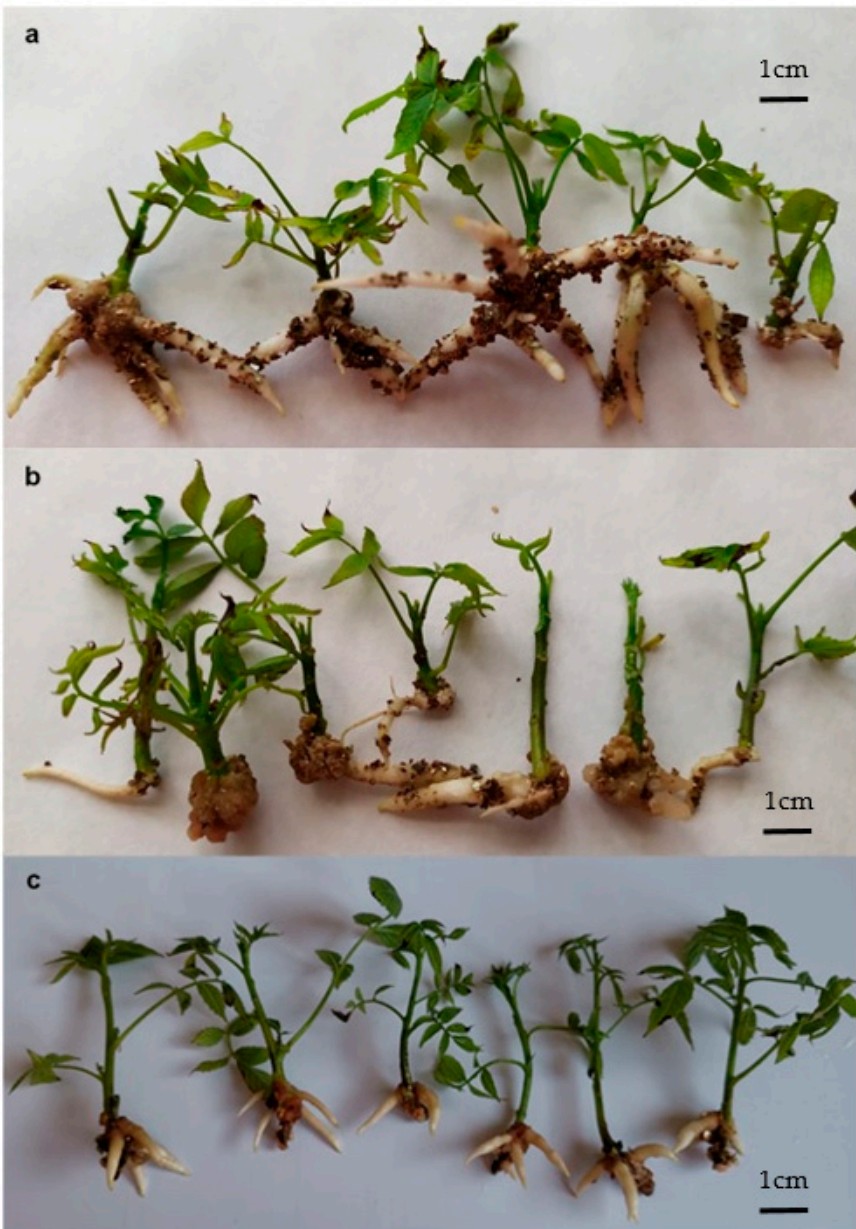

**Figure 3.** 'Form 3' rooted microshoots in DKWC using FeEDDHA as iron source (**a**,**c**) and MS culture media (**b**) in laboratories 1 (**a**,**b**) and 2 (**c**) before transplanting.

Unlike the culture medium, IBA is used almost unanimously for walnut root induction, although variations can be found regarding the more suitable concentrations. Thus, for Persian walnut variety 'Serr' registered an important reduction of rooting using concentrations above 19 μM in $\frac{1}{2}$ of DKW's macronutrients [57]. Whereas, Docet-Sanjuan et al. [58] concluded that for rooting of hybrid 'A69' (*J. nigra* $^x$ *J. regia*) the optimum level of IBA might be localized between 32 and 100 μM also using DKW formulation but reducing macronutrients four times. For all four genotypes used in this research, DKWC supplemented with 50 μM of IBA has offered the best conditions for rooting. Although since dramatic differences have not been registered regarding rooting percentage, it would be recommendable to continue assessing both formulations, investigating their interaction with other factors, such as IBA concentration.

Similar to proliferation, there was no interaction of rooting with the execution place (Table 1), although the percentage of microshoots forming roots and the number of roots were higher for all genotypes in laboratory 1 than in 2 (Table 2). However, the quality

of rooted microshoots was superior in laboratory 2 (Figure 3c) at the end of expression stage, bearing greener leaves than those from laboratory 1, and without any sign of wilting and defoliation (Figure 3a). It might suggest a differential exhaustion of culture medium components between laboratories, since higher rooting percentages and number of roots per microshoots were obtained in laboratory 1. Some other authors [34] have also observed that using the same culture medium and conditions, microshoots from vessels with more explants inoculated stopped growing earlier than those with lower quantities, appearing a general declination of microshoots, i.e., wilting and defoliation, by the likely exhaustion of culture medium components.

### 3.4. Acclimatization

The conditions here described seem suitable for the weaning of rooted microshoots, promoting the fast growing of roots, and favoring the survival of produced plants. Hence, at the end of the first step of hardening, a profuse root formation was observed, emerging throughout the Jiffy's covers for most of plants (Figure 4a). In general, low mortalities were registered for all genotypes and treatments, ranging from 8.9 to 50.9%, being 'Liaohe-1' the clone with the highest average survival (81.1%). These are outstanding results, comparable, even better, to those registered for different walnut genotypes. Thus, Voyiatzis and McGranahan [59] increased up to 66.5% the survival rate of walnut clone TRS using a latex antitranspirant. Hackett et al. [60] had to design a complex system with controlled humidity to reduce to 19% the losses during hardening of ex vitro rooted plants from 40 walnut rootstocks. Whereas, with the same protocol using DKWC formulation, for nine clones of Mj209xRa values of mortality between 16.7 and 67.3% were registered [17]. This confirms that the protocol here executed, under the described conditions, would allow the commercial micropropagation these genotypes.

Nevertheless, survival was significantly different amongst genotypes; although, the culture medium was the main variation factor (Tables 1 and 2). Thus, all rooted microshoots from DKWC had a greater ability to overcome ex vitro conditions than those from MS, which might agree with the state of plants in the moment of moving to ex vitro acclimatization. Here, a differential degree of defoliation at the end of root expression was observed, depending of the origin of microshoots, with greater affectations for those that grew in culture medium MS than in DKWC. Both McGranahan et al. [52] and Licea-Moreno [34] have observed that defoliated rooted-microshoots have less possibilities to overcome ex vitro hardening. Regardless of the differences caused for culture media once the plants were acclimated, the phenotypic variations almost disappeared as can be observed in Figure 4b,c.

Since it has been suggested that most of roots found during the first stages of acclimatization are in vitro formed [35], the ability to root is another factor with a likely great influence on survival. It may explain why higher mortality was registered for rooted microshoots from MS regarding those from DKWC (Table 2). This was confirmed once a direct correlation was calculated between the number of formed roots and survival (Spearman coefficient $r = 0.725$, $p < 0.0001$, n = 162). Additionally, it was also determined that the mortality of microshoots with 1 and 2 roots was considerably higher (84.1%) than that registered for those with 3–6 roots (7.6%). Previously, Chenevard et al. [61] associated the improvement of survival of microshoots from hybrid *J. nigra* n° 23 × *J. regia* to certain morphological characters, as the number of adventitious roots formed. For Mj209xRa hybrid clones a likely relationship between rooting and survival has also been suggested [17]. Therefore, the number of roots formed might be used as marker of the acclimatization success, and an internal control of micropropagation protocol, along with a multiplication rates and the quality of micropropagated plants.

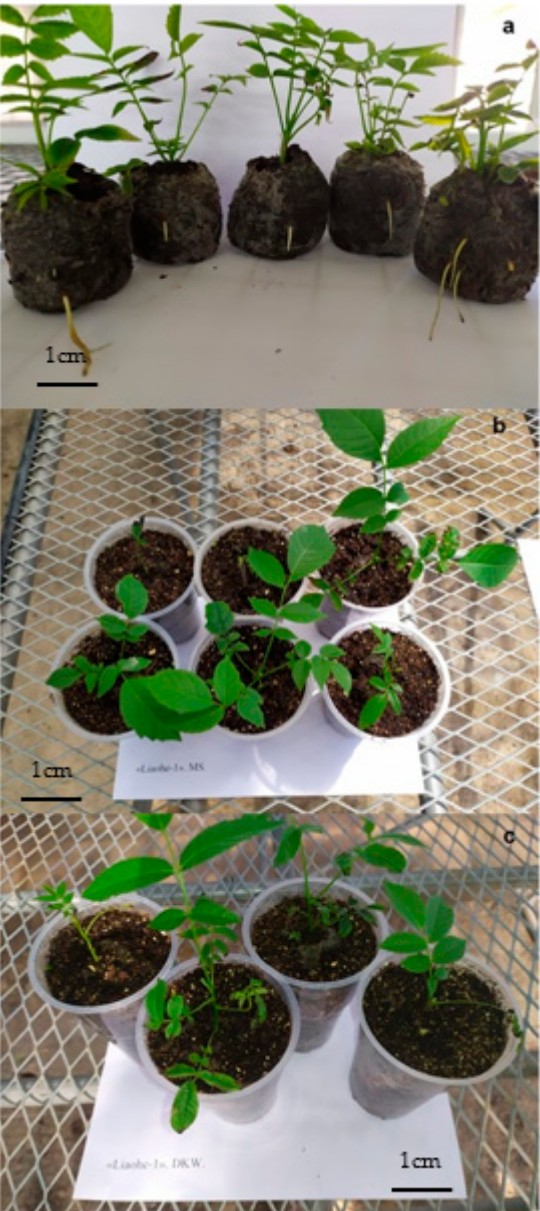

**Figure 4.** Acclimatization to ex vitro conditions of rooted microshoots obtained in laboratory 1 from clone Liaohe-1. (**a**) Micropropagated plants after 3 weeks of initiation of hardening, ready for the second transplant: the 3 plants on the left were micropropagated with DKWC, the other 2 on the right with MS. Autonomous plants from MS (**b**) and DKWC (**c**) culture media, at the end of acclimatization stage.

For the final assessment of the proposed protocol, the genetic profile of acclimated plants of four clones were compared with their corresponding donor trees. The five microsatellite loci used were selected on the basis of their reasonable high polymorphism registered for these genotypes. Since in vitro culture has been reported as a source of genetic variations, the analysis of the genetic stability is key for the validation of protocol. However, no discrepancies (Figure 5) were detected between the cloned materials and the original trees, maintaining the same correspondence of allelic sizes among both groups. This suggests that the proposed micropropagation protocol does not generate genotypic variations, at least for the assessed loci. The same results were registered for clones from Mj209xRa progeny using a similar micropropagation protocol [17].

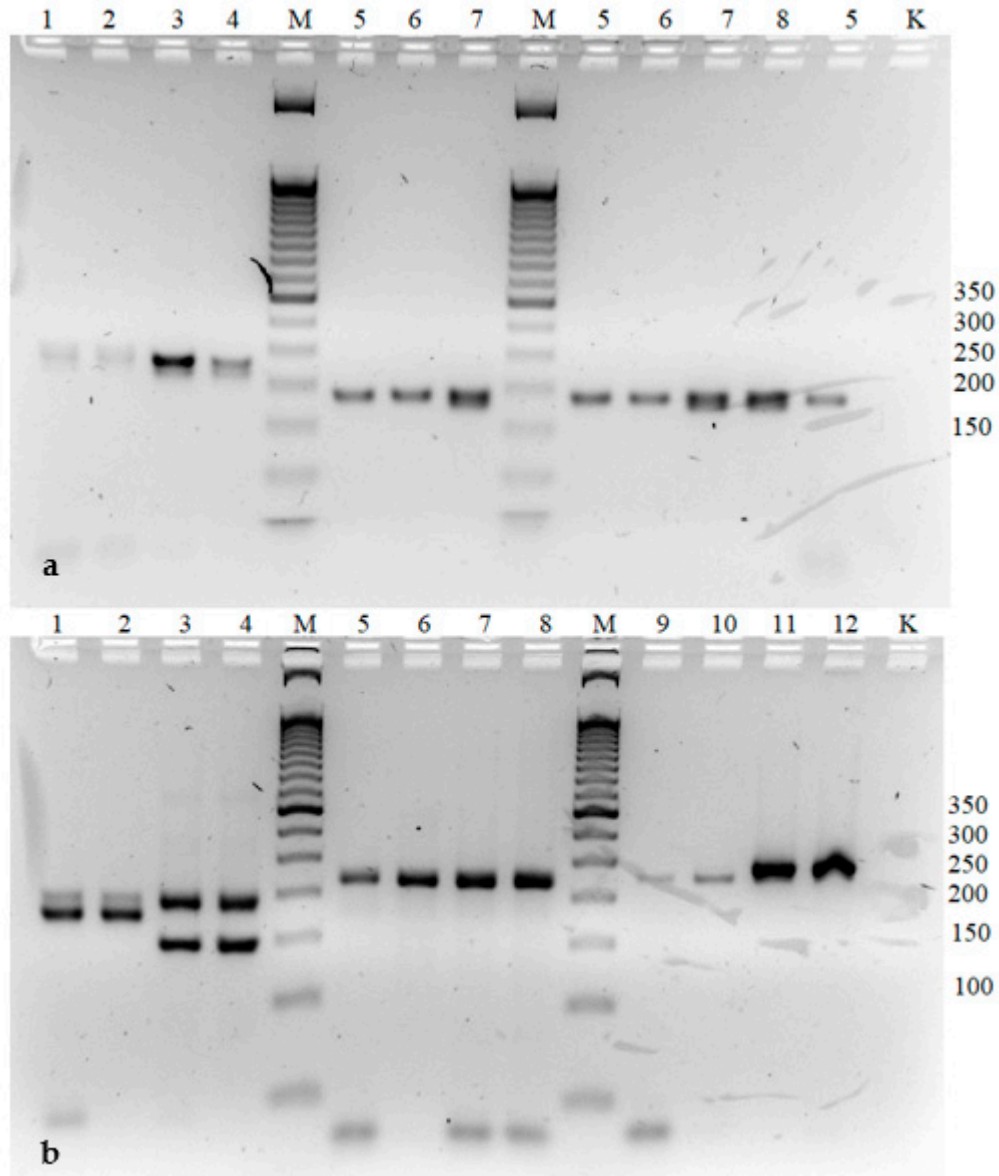

**Figure 5.** Electropherograms for 'Form 4' and 'Liaohe-1' donor trees and their clones for 5 SSR loci. (**a**) PCR amplifications for loci WGA009 (columns 1–4) and WGA001 (columns 5–8) for donor trees of 'Form 4' (columns 1, 5) and 'Liaohe-1' (columns 3, 7), and their corresponding clones (columns 2, 4, 6, 8). (**b**) PCR amplifications for loci WGA276 (columns 1–4), WGA089 (columns 5–8), and WGA321 (columns 9–12) for donor trees of 'Form 4' (columns 1, 5, 9) and 'Liaohe-1' (columns 3, 7, 11), and their corresponding clones (columns 2, 4, 6, 8, 10, 12). M, 50 bp DNA ladder. K, negative PCR control.

## 4. Conclusions

The results presented here allowed to deepen understanding of some critical aspects of walnut micropropagation, that determined the success of in vitro establishment, the growth and the proliferation, the rooting ability and the survival of four Persian walnut varieties. Besides, for the first time an approach to the assessment of transferability and reproducibility of a micropropagation protocol in two different locations was presented.

Thus, during in vitro establishment it was determined that besides genetic factors, the culture medium and the collection season had a great influence on microbial contaminations and the rate of phenolic releasing, with a direct repercussion on the percentage of established explants. It was demonstrated that the selection of season for initiation has a great repercussion on the obtained results.

For proliferation, none of the original formulations were able to sustain the growing of genotypes. Therefore, the substitution of FeEDTA by FeEDDHA, and the introduction of PG were necessary for the reproduction of four clones, with the best multiplication rates obtained with DKWC-m culture medium.

Rhizogenic ability was also improved in DKWC formulation regarding MS, being especially promoted the emission of more roots per microshoot. Similarly, microshoots from DKWC formulation registered the highest survival. These results undoubtedly may be the baseline for the commercial scaling up of micropropagation of these promising genotypes, once the multiplication rates, the rooting percentages and the reduced mortality during weaning were comparable to those obtained for other genotypes and laboratories.

The findings of the lack of genetic discrepancies for the six loci used between micropropagated plants and their corresponding donor trees, support the strength of the proposed micropropagation protocol.

Reproducibility is an axiomatic definition applied to most of micropropagation protocols, and frequently underestimated for its repercussion on the success of transference. As many factors, some of them unknown and/or uncontrollable, might determine the results of transference, this task was not aimed at determining the individual contribution of each one to the obtained results, but its reproducibility degree. Thus, it was demonstrated that the same protocol used in two different laboratories could be reproduced to some extent, although significant variations were registered. This highlights the importance to reproduce as close as it is possible the conditions described for each protocol.

**Author Contributions:** A.K.A. and S.N.O. proposed the rationale of research and provided plant materials. T.K.Y., S.G.-G. and R.J.L.-M. planned the in vitro experiments. T.K.Y., S.G.-G. and T.V.Y. executed the in vitro experiments. M.K. and R.J.L.-M. planned the SSR assessment. M.K. conducted PCR amplifications and determined allelic profiles. R.J.L.-M. performed statistical analysis and prepared the draft. All authors reviewed and approved the final version of paper. All authors have read and agreed to the published version of the manuscript.

**Funding:** This research was executed under the project AP 05134783, entitled "Accelerated methods of creating a collection of highly adaptive varieties and forms of walnut (*Juglans regia* L.) in Kazakhstan", granted to Issyk State Dendrology Park and supported by the Science Committee of the Ministry of Education and Science of the Republic of Kazakhstan.

**Institutional Review Board Statement:** Not applicable.

**Informed Consent Statement:** Not applicable.

**Data Availability Statement:** Data is contained within the article.

**Acknowledgments:** Special thanks to Femke Gouweleeuw-Hoogervorst, for her critical comments and suggestions, and Luis Grimal, General Administrator of WalnutRD, for his kind collaboration in the performing of this research.

**Conflicts of Interest:** The authors declare no conflict of interest. The funders had no role in the design of the study; in the collection, analyses, or interpretation of data; in the writing of the manuscript, or in the decision to publish the results.

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
