# Peer review of "Unraveling Factors Affecting Micropropagation of Four Persian Walnut Varieties"

_agronomy, doi:10.3390/agronomy11071417_

Round 1
Reviewer 1 Report
The article on “Unraveling factors affecting micropropagation of 4 Persian walnut varieties” provides a valuable evaluation of the effects of two different tissue culture media on micropropagation of four different walnut varieties. The research is within the scope of the journal and it is a valuable contribution to walnut propagation. However, the paper would benefit from language editing services.
A few specific suggestions include:
Figure 2: The figure caption could provide greater detail, and a description of the individual images (a-d) should be provided for each image. The variety should also be listed.
Tables 2 and 3 Titles: The lab name should be consistent with what is used throughout the paper (ex: lab 1 instead of ISDP and lab 2 instead of Walnut RD)
Table 2: Only the data from one lab is included (lab 1), but including both labs, and the means for both for comparison would be recommended.
Table 3: Only one media (DKWC-m) and one lab (lab 2) are included, but including the other media (MS-m) and lab 1 along with the overall means would be recommended.
Figure 3: Variety should be listed. The MS-m for lab 2 is not included. The figure caption could be included to provide a description of each individual image.
Figure 4: Information about media is not included for a) and information about which lab is also not included.
A table on the percent survival by variety, media formula, lab, and collection season would be a helpful addition.
The collection season (February or May) is evaluated for initial establishment of cultures only. However, this factor would be interesting to examine at later stages and for percent survival.
Author Response
The article on “Unraveling factors affecting micropropagation of 4 Persian walnut varieties” provides a valuable evaluation of the effects of two different tissue culture media on micropropagation of four different walnut varieties. The research is within the scope of the journal and it is a valuable contribution to walnut propagation. However, the paper would benefit from language editing services.
A few specific suggestions include:
Figure 2: The figure caption could provide greater detail, and a description of the individual images (a-d) should be provided for each image. The variety should also be listed.
Fixed. Instead the caption “Proliferation in formulation DKWC (a) replacing FeEDTA by FeEDDHA (b), and supplementing culture media with Phloroglucinol (c) in laboratories 1 (a-c) and 2 (d)”, a new one more explicative title has been proposed “Microshoots in the stage II (proliferation) cultivated in laboratories 1 (a-c) and 2 (d) with the corrected formulation of DKWC [19]. (a) Microshoots cultured in the DKWC formulation using FeEDTA. (b) Microshoots cultured in the DKWC formulation using FeEDDHA. (c, d) Microshoots cultured in the modified formulation of DKWC (DKWC-m) supplemented with Phloroglucinol, and using FeEDDHA as iron source”
Tables 2 and 3 Titles: The lab name should be consistent with what is used throughout the paper (ex: lab 1 instead of ISDP and lab 2 instead of Walnut RD)
Attended, and amended.
Table 2: Only the data from one lab is included (lab 1), but including both labs, and the means for both for comparison would be recommended.
Attended, and amended. Tables 2 and 3 were merged
Table 3: Only one media (DKWC-m) and one lab (lab 2) are included, but including the other media (MS-m) and lab 1 along with the overall means would be recommended.
Laboratory 2 is highly experienced in walnut micropropagation. They used in its routine the DKWC-m formulation, since other culture media (included MS and DKWC) have not been effective. Once in laboratory 2, the same results were obtained, i. e. with DKWC-m, as a previous step to the scaling up of the best protocol, was decided to assess how the transference to a new location might affect the experimental results. Therefore, was considered that the evaluation of MS under laboratory 2 conditions would not provide relevant data.
Figure 3: Variety should be listed. The MS-m for lab 2 is not included. The figure caption could be included to provide a description of each individual image.
Attended, and amended.
Figure 4: Information about media is not included for a) and information about which lab is also not included.
Attended, and amended.
A table on the percent survival by variety, media formula, lab, and collection season would be a helpful addition.
In table 2 most of these data are provided. For laboratory 2, the acclimation of obtained vitroplants was not performed since previous results have allowed to know that in vitro protocol was able to produce high quality rooted-microshoots, able to overcome the weaning. Besides, in laboratory 2 has been established some standards criteria to determine whether a rooted microshoots might survive to hardening, which have sustained its commercial production of different walnut species and genotypes for 2 years.
The collection season (February or May) is evaluated for initial establishment of cultures only. However, this factor would be interesting to examine at later stages and for percent survival.
Certainly, it may be worth to know if the starting season has some influence on survival, as many other factors that have not been included in this research. However, similarly to other laboratories, the microshoots produced in laboratory 1 fits to the standards of quality needed to survive during acclimation, as has been showed in this research.
Reviewer 2 Report
The paper is an interesting study on micropropagation of four walnut varieties. The composition of basal medium, growth regulators and age of explants have been investigated as important factors for successful micropropagation and the reproducibility of the proposed technique was confirmed by two laboratories, locate in different geographical locations. In general, the paper is interesting and well organized. I would like to make the following remarks:
- Replace “4” with “four” in title
- Figure 1 does not show data for all four investigated varieties;
- There is no figure or table that showed the lack of genetic variabilities. Here, a picture of agarose gel showing the SSR mapping would be helpful;
- Please check spelling for example “…might constitute the basement for…”
Author Response
The paper is an interesting study on micropropagation of four walnut varieties. The composition of basal medium, growth regulators and age of explants have been investigated as important factors for successful micropropagation and the reproducibility of the proposed technique was confirmed by two laboratories, locate in different geographical locations. In general, the paper is interesting and well organized. I would like to make the following remarks:
- Replace “4” with “four” in title
Attended, and amended.
- Figure 1 does not show data for all four investigated varieties;
In vitro introduction is highly dependent of the availability of starting materials. Thus, some genotypes were not able to produce enough sticks/shoots for all experiments. For this research, only treatments with at least 10 explants were included. That’s why the results for some genotypes were not included. Anyway, this annotation was included in MM.
- There is no figure or table that showed the lack of genetic variabilities. Here, a picture of agarose gel showing the SSR mapping would be helpful;
Attended, and amended. Two electropherograms have been provided in Figure 5 showing the maintaining of identity of micropropagated clones for 5 microsatellite loci. Since, extra works have been necessary, modifications in authorship have also been included.
- Please check spelling for example “…might constitute the basement for…”
Attended, and amended. Additionally, all draft was checked again, and corrected by native English-speaker
Reviewer 3 Report
Congratulation, a complex and well-documented article. Also interesting are the results obtained in the two laboratories. All the best.Author Response
Congratulation, a complex and well-documented article. Also interesting are the results obtained in the two laboratories. All the best.
Thanks for your comments
Reviewer 4 Report
Dear Authors,
The manuscript is well-structured, has a great flow of ideas, and brings important information concerning the issues of the micropropagation of walnut.
Obs:
In abstract
change to "4 Persian walnut " with "4 Persian walnut (Juglans regia L.) "
Author Response
Dear Authors,
The manuscript is well-structured, has a great flow of ideas, and brings important information concerning the issues of the micropropagation of walnut.
Obs:
In abstract
change to "4 Persian walnut " with "4 Persian walnut (Juglans regia L.) "
Attended, and amended.